# Inferring a network from dynamical signals at its nodes

**Corey Weistuch**[1,2], **Luca Agozzino**[1], **Lilianne R. Mujica-Parodi**[1,3,4,5,6], **Ken A. Dill**[1,3,7] *

**1** Laufer Center for Physical and Quantitative Biology, Stony Brook University, Stony Brook, New York, USA, **2** Department of Applied Mathematics and Statistics, Stony Brook University, Stony Brook, New York, USA, **3** Department of Physics and Astronomy, Stony Brook University, Stony Brook, New York, USA, **4** Department of Biomedical Engineering, Stony Brook University, Stony Brook, New York, USA, **5** Program in Neuroscience, Stony Brook University, Stony Brook, New York, USA, **6** Athinoula A. Martinos Center for Biomedical Imaging, Massachusetts General Hospital and Harvard Medical School, Boston, Massachusetts, USA, **7** Department of Chemistry, Stony Brook University, Stony Brook, New York, USA

* dill@laufercenter.org

## Abstract

We give an approximate solution to the difficult inverse problem of inferring the topology of an unknown network from given time-dependent signals at the nodes. For example, we measure signals from individual neurons in the brain, and infer how they are inter-connected. We use Maximum Caliber as an inference principle. The combinatorial challenge of high-dimensional data is handled using two different approximations to the pairwise couplings. We show two proofs of principle: in a nonlinear genetic toggle switch circuit, and in a toy neural network.

**Data Availability Statement:** All code files are available on https://github.com/Corey651/MaxCal_Network.

**Funding:** The research was funded by the WM Keck Foundation (LRMP, KAD), the NSF BRAIN

## Author summary

Of major scientific interest are networks—the internet, commercial supply chains, social media, traffic, biochemical reactions inside cells, the neurons in the brain, and many others. Often, the challenge is to measure a few rates at a limited number of nodes of the network, and to try to infer more information about a complex network and its flow patterns under different conditions. Here we devise a mathematical method to infer the dynamics of such networks, given only limited experimental information. The tool best suited for this purpose is the Principle of Maximum Caliber, but it also requires that we handle the challenge of the high-dimensionality of real-world nets. We give two levels of approximation that reduce this to the simpler problem of inferring the dynamics of each node individually. We show that these approximations provide novel insights and accurate inferences and are promising for drawing inferences about large-scale biophysical and other networks.

This is a *PLOS Computational Biology* Methods paper.

Initiative (LRMP: ECCS1533257), the NSF BRAIN
Initiative (LRMP, KAD: NCS-FR 1926781) and the
Stony Brook University Laufer Center for Physical
and Quantitative Biology (KAD). The funders had
no role in the study design, data collection and
analysis, decision to publish, or preparation of the
manuscript.

**Competing interests:** The authors have declared
that no competing interests exist.

# Introduction

## Learning the properties of a network from measurements at its nodes

We are interested in solving the following 'inverse problem': you measure time-dependent signals from individual agents. Those agents behave in a correlated way. That is, they are connected in some network that is unknown to you. The goal is to infer the interactions between these agents from their correlations. For example, measure the protein concentrations that are produced from an unknown gene network, and infer the degree to which the proteins activate or inhibit each other. Or measure the firings of individual neurons and infer the neuron-neuron connection strengths in the brain. These problems are the 'inverse' compared to the common situation of knowing a network and computing the flows through it.

While there are many powerful techniques for inferring which nodes are linked and how strongly, we are interested here in inferring the propagation dynamics distributions [1–5]. That is, we seek to infer a model, or a probability distribution, for how the activities of our network of agents evolve over time. In contrast to the assumptions of common Bayesian approaches to this problem, we rarely know the shape or the structure of this model [6, 7]. Instead, we are given limited information and seek to infer the model directly from the data itself. The method of choice for inferring dynamical processes from limited information is the *Principle of Maximum Caliber* (Max Cal) [8–14]. Max Cal is a procedure that predicts full stochastic distributions by maximizing the route entropy subject to the constraint of a few known average rates. Thus, Max Cal is to dynamics what Maximum Entropy inference (Max Ent) is to equilibrium. Like Max Ent, Max Cal requires minimal model assumptions that are not warranted by the data itself. For example, Max Cal has proven capable of reproducing many known results from non-equilibrium physics, such as Fick's law and the master equation [14–16]. In addition, Max Cal has been show to accurately predict single-cell dynamics [17, 18], such as in gene circuits [19–21] and and stochastic cycles [22, 23], directly from noisy experimental data.

The challenge here is that the number of possible interactions (here the node-node couplings) grows rapidly with system size (the number of nodes in the network and length of time of observing the signals). So, direct implementation of Max Cal is limited to small or simplified systems [24–27]. For larger and more realistic situations, this Max Cal inference procedure becomes computationally intractable. In other matters of physics, the dimensionality of the problem is reduced by various approximations, including *variational approximation* and *perturbation theory* [28–31]. These techniques have been used to reduce the dimensionality in other high-dimensional inference problems [32–43].

Here, we adapt such methods for inferring high-dimensional, heterogeneous dynamical interrelationships from limited information. Related generalizations have been previously used to infer dynamical interactions in continuous-time Markovian networks [44, 45]. However, these approaches make strong assumptions either about the dynamics or about unknown transition rates. Here instead, with Max Cal, we can infer both the dynamics and interactions within arbitrarily complex, non-equilibrium systems, albeit in an approximate way. We describe two different levels of approximation: Uncoupled and Linear Coupling.

## The problem

Suppose you run an experiment and record the activity of $N$ arbitrarily interacting agents (the nodes, $i = 1, 2, \ldots, N$ of a network) over some time $T$ (see for example Fig 1). The data arrives as a time series: $\mathbf{v}(t) = \{v_1(t), v_2(t), \ldots, v_i(t), \ldots, v_N(t)\}$, also called a trajectory, $\Gamma$ (from $t = 0$ to $T$). From the signals, we aim to predict the coupling strengths between the nodes. Our model

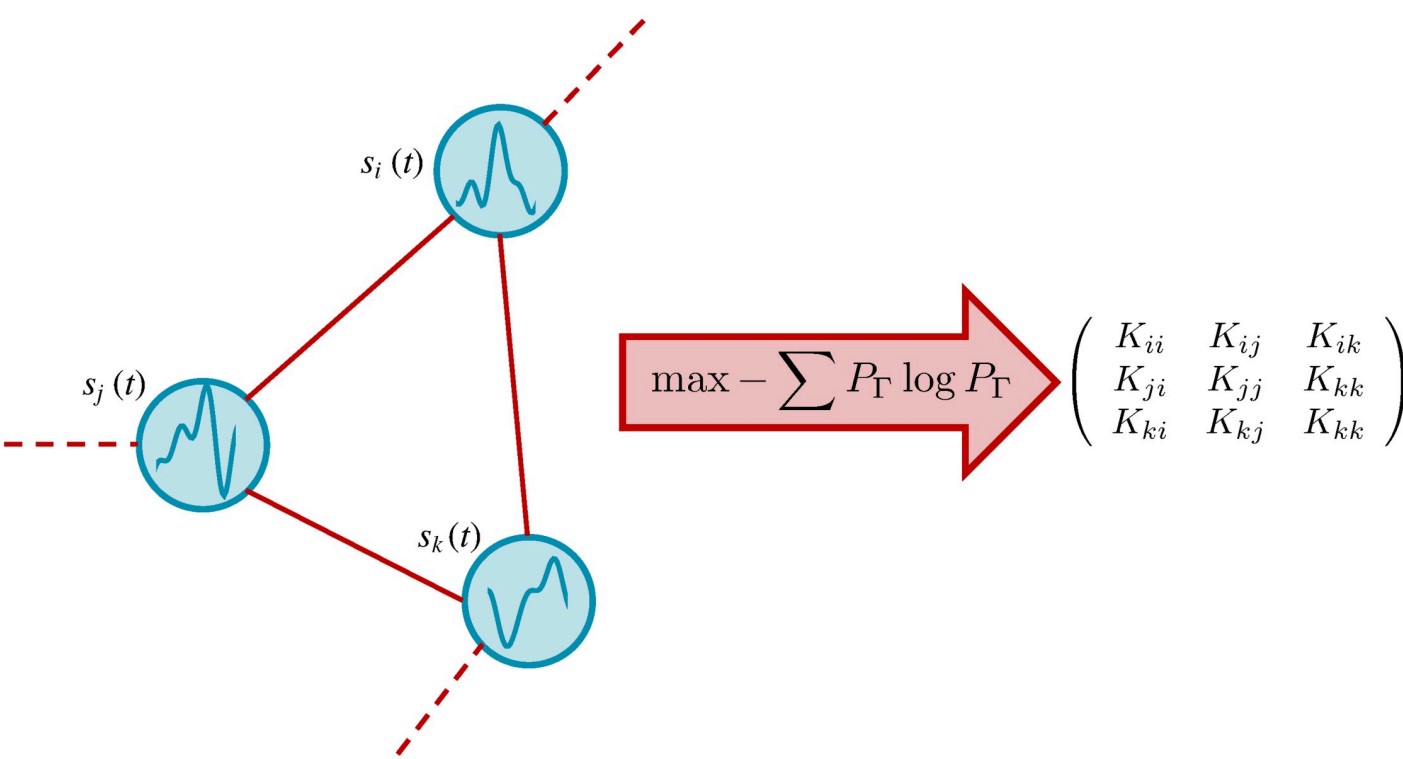

**Fig 1. Maximum Caliber (Max Cal) infers network structures.** From time-dependent signals from nodes (left) we maximize the path entropy, or *Caliber*, to infer the interaction strength (structure) $K_{ij}$ between edges $i$ and $j$.

should reliably predict certain averages over the data, with otherwise the least possible bias. Such problems are the purview of the principle of Maximum Entropy or its dynamical extension, *Maximum Caliber* (Max Cal) [8–13]. The principle of Max Cal chooses the unique distribution, $P_\Gamma$, that maximizes the path entropy, or *Caliber*, over all acceptable distributions $\{P_\Gamma\}$, while respecting the observed constraints. The constraints here are the mean value, $M_i(t)$ over all possible paths, and the correlations, $\chi_{ij}(t,s)$:

$$M_i(t) = \langle v_i(t) \rangle, \qquad \chi_{ij}(t,s) = \langle v_i(t) v_j(s) \rangle \tag{1}$$

for all times $t$ and $s$ over all agents $i$ and $j$. The Caliber is expressed as

$$\mathcal{C} = - \sum_\Gamma P_\Gamma \ln P_\Gamma + \mu \sum_\Gamma P_\Gamma + \sum_\Gamma \sum_{i,t} h_i(t) v_i(t) P_\Gamma$$
$$+ \sum_\Gamma \sum_{i,j,t,s} K_{i,j}(t,s) v_i(t) v_j(s) P_\Gamma \tag{2}$$

where the sum over $\Gamma$ is a sum over all the possible realizations of the full time series. Here $h_i(t)$ and $K_{i,j}(t,s)$ are the Lagrange multipliers that each enforce the constraints in Eq 1. The other Lagrange multiplier, $\mu$, ensures the distribution is normalized (the probabilities sum to one).

Maximizing the Caliber with respect to all possible distributions $\{P_\Gamma\}$ gives

$$P_\Gamma = \frac{1}{Z} \exp \left[ \sum_{i,t} \left( h_i(t) + \frac{1}{2} \sum_{j,s} K_{ij}(t,s) v_j(s) \right) v_i(t) \right] \tag{3}$$

$Z$ is the *dynamical partition function*, the quantity that normalizes $P_\Gamma$. By analogy with the Ising model for equilibrium systems, $h_i(t)$ represents the strength of the external fields to which the system is coupled, whereas $K_{ij}(t, s)$ are the couplings between the components of the system.

## Results

### The Uncoupled Max Cal Approximation

We aim to compute $h_i$ and $K_{ij}$ for every time point. This presents a combinatorial challenge for large networks or long trajectories. We describe two levels of approximation to overcome this challenge. In the present section, we describe our simplest approximation, representing a *mean-field* or *Uncoupled* approach, which allows us to solve even large systems [39, 45]. This method works by breaking the full inference problem into simpler, independent subproblems. For our application, this suggests that we try uncoupling the trajectories of each object ($i$) which we denote $\Gamma_i$. The approximate trajectory distribution $Q_\Gamma$ then factorizes into the product:

$$Q_\Gamma = \prod_i Q_{\Gamma_i} \tag{4}$$

Eq 3 shows that this approximation is exact when all of the coupling coefficients $K_{ij}(t, s)$, $i \neq j$, are 0. We can force this condition by temporarily ignoring all pairwise constraints corresponding to $K_{ij}(t, s)$ and satisfying the remaining, $i = j$, Max Cal constraints (from Eq 1). The now Uncoupled distributions are given by:

$$Q_{\Gamma_i} = \frac{1}{\tilde{Z}_i} \exp\left[ \sum_t \left( \tilde{h}_i(t) + \frac{1}{2} \sum_s \tilde{K}_{ii}(t, s) v_i(s) \right) v_i(t) \right] \tag{5}$$

This then gives a new set of effective Lagrange multipliers, $\tilde{h}_i(t)$ and $\tilde{K}_{ii}(t, s)$, which absorb the average effects of the neglected pairwise interactions.

In summary, this Uncoupling Approximation reduces the inference problem to solving independent single-node problems for each $i$. These single-node inference problems are readily solved [12, 25]. Clearly, however, this naive approximation fails to capture any pairwise correlations between agents ($i \neq j$). Instead, it is meant to be used when the fluctuations in the interactions between agents can be neglected. The following section describes a next better approximation, based on Linear Response Theory [35].

### The Linear Coupling Max Cal Approximation

Here, we go beyond the uncoupling assumption and take the first-order perturbation term around our Uncoupled Approximation. We call this the *Linear Coupling Max Cal* Approximation. The first-order approximation for the Lagrange multipliers for each agent $i$ are given by (see Methods, How to choose the Uncoupled distribution):

$$h'_i(t) = \tilde{h}_i(t) - \sum_{j \neq i} \sum_s K'_{ij}(t, s) M_j(s)$$

$$K'_{ii}(t, s) = \tilde{K}_{ii}(t, s) \tag{6}$$

Eq 6—analogous to familiar mean-field models in physics—attempts to recover the true Lagrange multipliers (with' denoting the Linear Coupling Approximation) from the effective, Uncoupled Lagrange multipliers (denoted by $\sim$) [39, 45]. Thus our only remaining unknowns

are the values of the pairwise couplings $K'_{ij}(t, s)$. For our first order approximation, the Linear Coupling estimates for these Lagrange multipliers have a closed-form solution (see Methods, Eq 16) given by:

$$K'_{ij}(t, s) = -(C^{-1})_{ij}(t, s), \quad i \neq j \tag{7}$$

where $C_{ij}(t, s) = \chi_{ij}(t, s) - M_i(t)M_j(s)$ is the matrix of covariances. Once these estimates are known, the remaining Lagrange multipliers are easily found from Eq 6.

Below, we give two examples to illustrate two points. First, we show that even the Uncoupled Approximation can give a fairly accurate closed-form solution for a network with nonlinear interactions. We show this for a genetic toggle switch [24, 46]. Second, we show how the Linear Coupling Approximation readily handles a high-dimensional heterogeneous system, namely a toy network of neurons, which is otherwise computationally intractable.

## Finding stable states in the genetic toggle switch

Collins *et al.* engineered a synthetic circuit into a single-celled organism called a *genetic toggle switch* [46]. It consists of two genes (A and B, blue and yellow in Fig 2a). Each gene produces a protein that inhibits the other. So, in analogy with an electrical toggle switch, either A is being produced while B is turned off, or vice versa. This network has previously been computationally simulated using Max Cal [24], so it provides a 'Ground Truth' for comparison with our approximation here. The present exercise shows that Uncoupled Max Cal, which can be solved analytically, gives an excellent approximation to the nonlinearity and the phase diagram in this known system. Importantly, beyond this proof of principle, Uncoupled Max Cal is readily applicable to bigger more complex systems.

Here, our input data takes the form of the stochastic numbers of protein molecules (obtained, for example, from fluorescence experiments [24, 25]), totaling $N_A(t)$ and $N_B(t)$ on nodes $A$ and $B$ at time $t$. Our trajectories are the counts of the numbers ($l_\alpha$ and $l_\beta$) of newly produced proteins of types $A$ and $B$ respectively over each new short time interval $\delta t$. From these

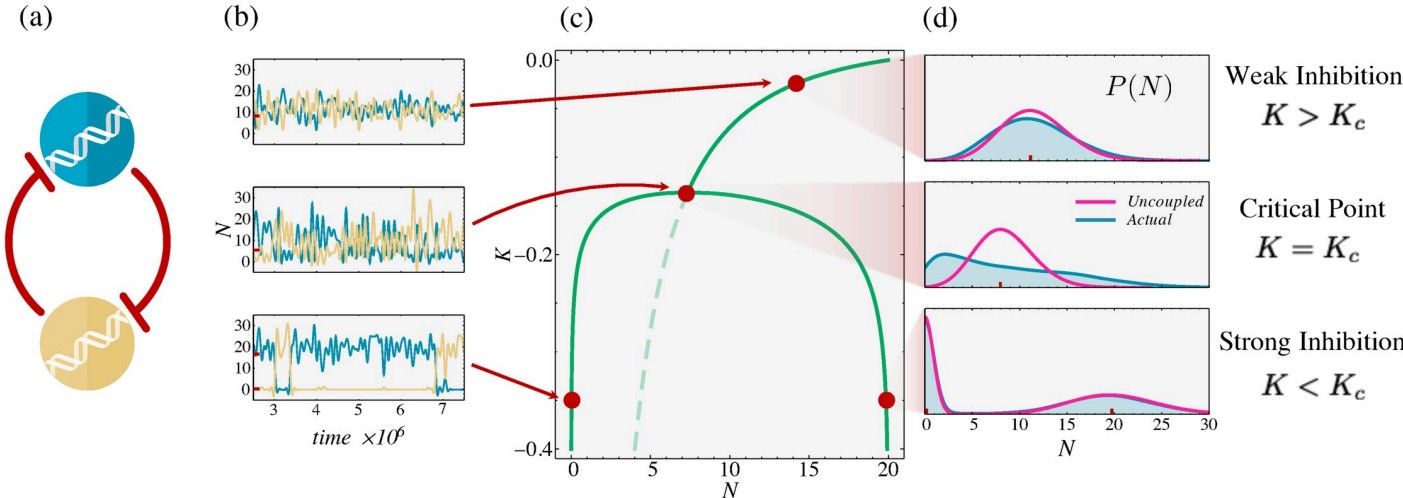

**Fig 2. The toggle switch 2-gene network.** (a) *A* and *B* inhibit each other. (b) Time trajectories of protein number $N_A$ (blue) and $N_B$ (yellow), simulated from the Ground Truth (Eq 24). (c) Its phase diagram, from the Uncoupled Max Cal Approximation. (green). Solid green: stable phases. Dashed green: unstable phase. Red dots (top to bottom and corresponding to Ground Truth simulations): Supercritical, single state ($K > K_c$). Critical point ($K_c$), big fluctuations. Subcritical, toggle switch, two bistable states ($K < K_c$). (d) The histograms of populations $P(N)$ in each case, comparing the Ground Truth (blue) to the Uncoupled result (magenta). The red lines correspond to the markers in the phase diagram. Uncoupled Max Cal captures the distribution correctly except at the critical point (center, truncated power law).

trajectories, we use Max Cal to compute three quantities: the production rate of each protein, the survival rates (the count of proteins that are not degraded), and the strength of the negative feedback. To keep the model simple, we suppose that both proteins have the same production rate, and both have the same survival rate. From the data, we obtain the average production and survival rates, $h_P$ and $h_S$, which are enforced in the Max Cal modeling as Lagrange multipliers. And, from the data, we obtain the correlation between production of A and survival of B, $\langle l_\alpha l_B \rangle$ (and vice-versa); these are enforced by a third Lagrange multiplier, $K$, the coupling coefficient [24] (see Methods, Toggle switch for details).

The behavior of this network is known from the Ground Truth simulations; see Fig 2b. There is a critical value, $K_c < 0$ of the coupling parameter (or repression strength). When the repression is weak ($K > K_c$), the circuit has a single stable state, producing equivalent amounts of $A$ and $B$ (top). Below the critical point, however, this circuit becomes a bistable toggle switch, either producing $A$ and inhibiting $B$ or vice versa (bottom). This transition corresponds to the bifurcation, from one to two stable points, in the phase diagram of the system (Fig 2c). While this phase diagram (green) is known from previous simulations, no analytical relationship was found, particularly for $K_c$, the critical point.

Here we have modeled this system using Uncoupled Max Cal (Eq 6) to find accurate (Fig 2c, green), analytical relationships for the phase diagram of the toggle switch (see Methods, Eqs 32 and 33). Away from the critical point, fluctuations in protein number have a minimal effect on the repression of our two genes. In other words, the production and degradation rates of each protein are approximately constant near each steady-state. As a result, Uncoupled Max Cal properly captures the full protein distributions away from the critical point (Fig 2d). At the critical point, however, the effects of these fluctuations are large and cannot be neglected, causing our Uncoupled Approximation to fail (Fig 2d, middle). Nevertheless, Uncoupled Max Cal allows us to calculate analytically the correct critical point (see Methods, Eq 34):

$$K_c = -e^{1-h_P-h_S} \tag{8}$$

## Learning the dynamical wiring of a network of neurons

Here we consider a brain-like neural network problem to illustrate how Linear Coupling Max Cal can infer a large network from limited information. Consider a network of $N$ neurons ($N = 40$ here). Taking the stochastic signals from the neurons, we want to infer the neuronal connectivities, and activity patterns. We illustrate how Linear Coupling Max Cal handles this even when we don't measure signals of all of the neurons together.

At any given time $t$, a neuron either fires (+1) or is silent (−1) in a time interval $\delta t$. The state of each neuron $i$, $v_i(t)$, is dependent on both the present and past states of other connected neurons. We assume only limited information is available, namely the mean values and correlations, as in Eq 1. The probabilities of different activity patterns $\{v_1(t), v_2(t), \ldots, v_N(t)\}$ are computed using Eq 3). This model resembles an Ising model of heterogeneous agents, which is often found effective in capturing observed neural activity [27, 47, 48]. In the context of neural activity, $h_i(t)$ (bias) controls the probability that neuron $i$ fires, while $K_{ij}(t, s)$ (connection strength) controls the probability that two neurons ($i$ and $j$) fire together. The challenge here for learning the dynamics is the large number of neurons [27, 49].

We test our predictions against a biologically plausible Ground Truth model of this network [27, 47] using time-independent Lagrange multipliers $h_i(t) = h_0$ and $K_{ij}(t, s) = K_{ij}(\tau)$, with $\tau = |t - s|$ (Fig 3; see Methods, Neural Network for the parameters of the model). $h_0 \ll 0$ was

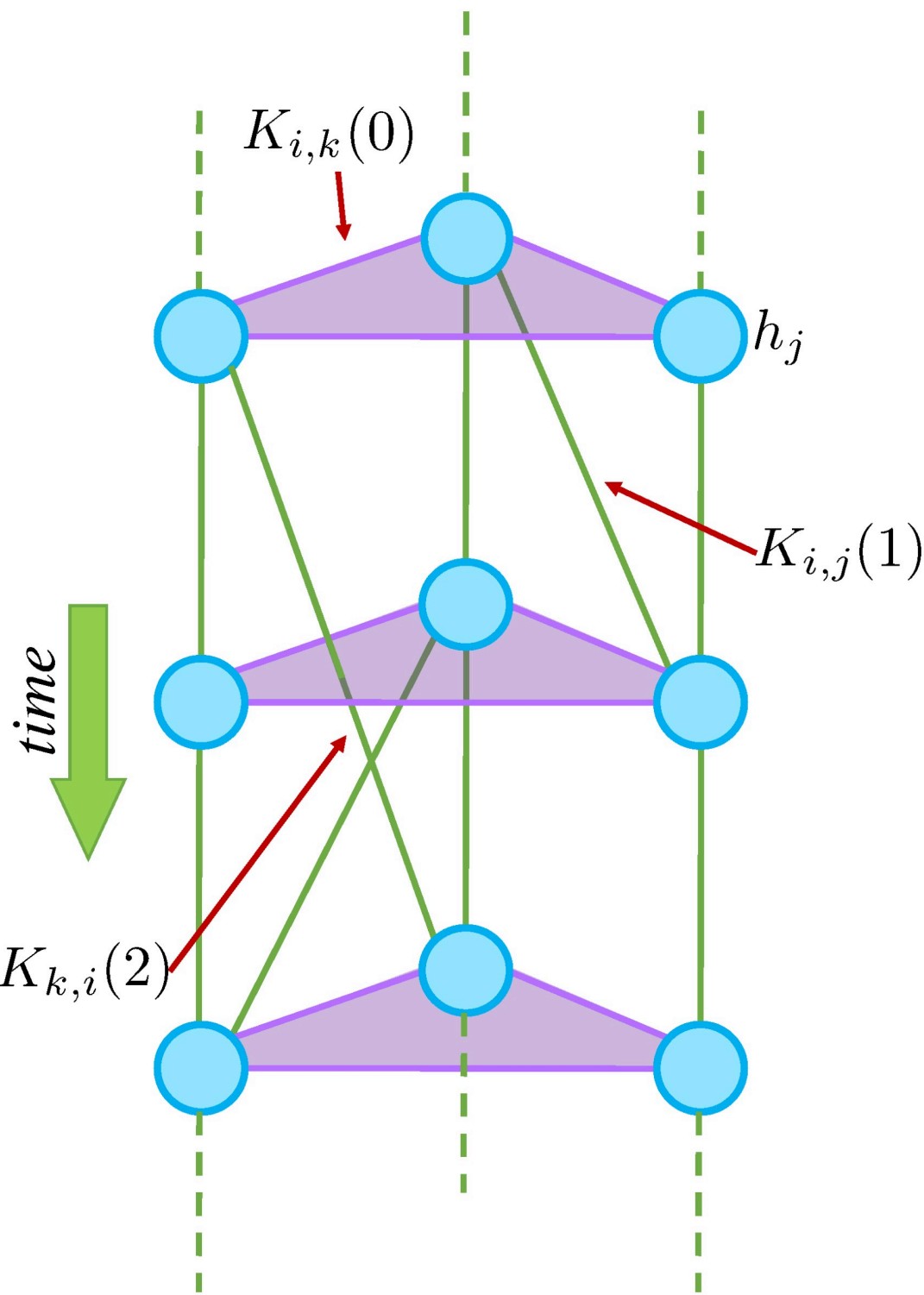

**Fig 3. Simplified neuron wiring diagram.** Blue circles are neurons each having bias $h_j$. green and purple edges are connections between neurons (signals separated by a time $\tau$) with strengths $K_{ij}(\tau)$.

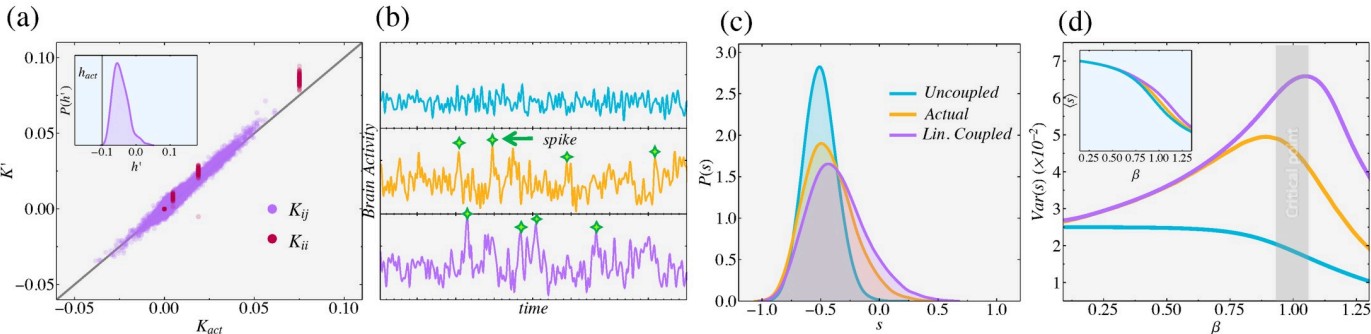

**Fig 4. The Neural network example.** (a) The Linear Coupling Approximation ($K'$ and $h'$) recovers neuron-neuron connection strengths ($K_{act}$) and biases ($h_{act}$, inset). The Uncoupled Approximation would estimate $K_{ij} = 0$. The black diagonal line represents perfect accuracy (b) Average neural activity (or synchrony, s) from the Uncoupled (blue), Linearly Coupled (purple), and true (orange) networks. Like the Ground Truth model, the Linearly Coupled model exhibits avalanches (spikes), an important feature of neural activity. (c) The histogram of $s$ for each model. The Linear Coupling model is much more accurate than the Uncoupled Approximation alone. (d). Model predictions for different connection strengths ($\beta$). (a-c) reflect $\beta = 1$, the critical point. While all methods capture mean activity $\langle s \rangle$, only the Linearly Coupled model captures the fluctuations $Var(s)$. (See Methods, Neural Network for the details of our Ground Truth as well as the implementation details).

chosen to reflect the tendency of real neurons towards silence, while $K_{ij}(\tau)$ was chosen from a normal distribution to reflect the heterogeneity between neurons [47]. A realistic assumption is that for $\tau > 3$, $K_{ij}(\tau) \approx 0$ [27]. In addition, although real neurons are usually silent, occasionally random firing of a few neurons can trigger a large cascade, or "avalanche" of activity [50]. These events can only occur when the wiring strengths between neurons (here our Ground Truth model) are tuned near a critical point, where the wiring strengths are weak enough to allow spontaneous neural activity but strong enough to force other neurons to entrain [47, 51].

Linear Coupling Max Cal (Eqs 6 and 7) recovers accurately the key features of neural activity present in the Ground Truth model (Fig 4). It requires input of only the means and correlations between the neurons (Eq 1). In sharp contrast to the Uncoupled Approximation $K_{ij}(\tau) = 0$, Linear Coupling Max Cal correctly recovers the dynamical connections between neurons (Fig 4a). We then took all three of these models and simulated (see Methods B) how average activity, or neural *synchrony*, $s(t) = \sum_i v_i(t)/N$ ($N = 40$) evolved over time [52]. In particular, the Linearly Coupled model correctly captures neural avalanches, where $s$ suddenly spikes and many neurons simultaneously fire, whereas the Uncoupled model does not (Fig 4b). It also correctly captures the spike frequencies (probabilities of $s > 0$; Fig 4c).

Linear Coupling Max Cal is just a first-order approximation, valid in the limit of weak interactions. Here, we also tested how this approximation errors changes as interactions are strengthened. Acting like an inverse temperature $\beta \sim T^{-1}$, we can modulate the average correlation strength between neurons by multiplying each Lagrange multiplier by $\beta$: $h_i \to \beta h_i$, $K_{ij} \to \beta K_{ij}$. When $\beta > 1$, connections are stronger; when $\beta < 1$, they are weaker. Fig 4d shows how well Linear Coupling Max Cal captures the features of neural synchrony, $P(s)$, over a wide range of $\beta$. As expected, both methods accurately capture the mean $\langle s \rangle$ value of synchrony, but only Linear Coupling reasonably captures the fluctuations, or variance $Var(s)$. In addition, the error is maximal near $\beta = 1$ (our original model), suggesting that our method gives reasonable results even in the worst-case (i.e. near critical points). Overall, the Linear Coupling Approximation provides fast, accurate estimates for the couplings within a large network ($N = 40$) of neurons that had previously been intractable [27, 47].

## Discussion

### When to use the different approximations

We have given two approximate methods for inferring stochastic network dynamics: the Uncoupled and Linear Coupling Max Cal methods. Here we describe when each method is relevant and how our approach might be improved upon.

Uncoupled Max Cal is useful when we are interested in identifying stable network configurations (such as steady-states in genetic circuits), along with the slow transitions between them, from limited experimental information. Here the method works when the interactions between agents are either very weak (and thus naturally uncoupled) or very strong. When interactions are strong, fluctuations away from these stable configurations are rare and can be neglected. Uncoupled Max Cal then infers the effective behavior of each agent near these stable configurations and, as in the genetic toggle switch (with two such configurations when $K < K_C$), adds them to reconstruct the full distribution of behaviors. For intermediate interactions, the classical Ginzburg-Landau theory of phase transitions can be used to identify when the critical points of a model can be predicted using the Uncoupled Approximation [53]. Thus, all these situations are cases when the fluctuations of the system are small.

Linear Coupling Max Cal is useful when fluctuations (i.e. cross-correlations) cannot be neglected. Akin to similar equilibrium approaches, this method is particularly useful when the correlations between agents are weak (see Methods, Quantifying the accuracy of Linear Coupling). However, just like for the Uncoupled Approximation, this method also works if the mean and correlation constraints are calculated when the network is fluctuating around a particular steady-state (such as the on/off configurations in the toggle switch).

Higher-order approximations can also be treated, as follows. We could employ the *Plefka expansion*, which has been fruitfully applied to equilibria [38]. Another option would be the *Bethe approximation*, starting from two-body, rather than one-body terms [41–43]. More generally, *mean-field variational inference* can be used to constrain arbitrary marginal and joint distributions [39, 54, 55], rather than means and variances. And deep learning methods could be used to learn higher-order interactions [35, 56, 57].

### Conclusions

We describe here a way to infer how the dynamics on multi-node networks evolves over time. We use an inference principle for dynamics and networks called Maximum Caliber [12–14]. Unlike previous methods, this approach utilizes only the available experimental data and requires minimal assumptions [1–6, 58]. Here, the direct interactions between nodes in a network are expressed in the couplings $K$. To solve the challenging problem of inferring these coupling from data, we introduce two levels of approximation—Uncoupling and Linear Coupling, which can render computations feasible even for networks that are large or have nonlinearities and feedback. While our method assumes knowledge of the relevant constraints and variables, one can directly leverage the strategies employed by previous applications of Maximum Caliber. In sum, the present approach is simple and computationally efficient.

## Methods

### How to choose the Uncoupled distribution

To approximate the true Max Cal distribution, $P_\Gamma$ using our Uncoupling approach, we restrict the maximization of Caliber to the set of factorizable distributions $Q_\Gamma$ (Eq 4). In particular, we

can easily solve Max Cal problems without interactions, so we choose $Q_{\Gamma_i}$ such that:

$$Q_{\Gamma_i} = \frac{1}{\tilde{Z}_i} \exp\left[\sum_t [\tilde{h}_i(t) + \frac{1}{2}\sum_s \tilde{K}_{ii}(t,s)v_i(s)]v_i(t)\right] \tag{9}$$

Here we discuss how to choose which $Q_{\Gamma}$, i.e. which values of $\tilde{h}_i(t)$ and $\tilde{K}_{ii}(t,s)$ to use as our approximation. Logically we want $Q_{\Gamma}$ to be as close to $P_{\Gamma}$ as possible. A common way to quantify this "distance" between probability distributions is the Kullback-Leibler (KL) divergence [59]:

$$D_{\mathrm{KL}}(P_{\Gamma}\,||Q_{\Gamma}) = \sum_{\Gamma} P_{\Gamma}\,\ln\,\frac{P_{\Gamma}}{Q_{\Gamma}} \quad or \quad D_{\mathrm{KL}}(Q_{\Gamma}\,||P_{\Gamma}) = \sum_{\Gamma} Q_{\Gamma}\,\ln\,\frac{Q_{\Gamma}}{P_{\Gamma}} \tag{10}$$

Notice, however, that the KL divergence is asymmetric; each choice gives a different optimal $Q_{\Gamma}$ with different advantages (see Methods, Minimization of KL divergences). Minimizing the forward divergence (left) implies choosing $Q_{\Gamma}$ that matches the one-body constraints, $M_i(t)$ and $\chi_{ii}(t,s)$, from our original Max Cal problem. Unfortunately, this choice also gives no clear relationship to the true Lagrange multipliers ($h_i(t)$, for example). Conversely, minimizing the reverse divergence (right) choice suggests that we choose $Q_{\Gamma}$ that satisfies our mean-field equation Eq 6, but the means in this equation are not guaranteed to relate to our experimental constraints. Intuitively, however, by uncoupling our agents, we aim to preserve their average dynamics (forward) by readjusting their external fields to compensate for the correlations that we are neglecting (reverse). Indeed, these solutions match up to first-order, allowing us to directly relate our easily solved Uncoupled Lagrange multipliers to their true values (see [38, 39, 60] for a proof and deeper insight).

## Minimization of KL divergences

Here we derive the dynamical mean-field equation Eq 6 by minimizing the KL divergences between the true Maximum Caliber distribution ($P_{\Gamma}$) and the Uncoupled Approximation ($Q_{\Gamma}$).

**Forward.**

$$
\begin{aligned}
F(\{\tilde{h}_i(t)\}, \{\tilde{K}_{ii}(t,s)\}) &= D_{\mathrm{KL}}(P_{\Gamma}\,||Q_{\Gamma}) \\
&= \langle \log\,P_{\Gamma}\rangle_P - \langle \log\,Q_{\Gamma}\rangle_P \\
&= \langle \log\,P_{\Gamma}\rangle_P - \sum_{i,t}\left[\tilde{h}_i(t)M_i(t) + \frac{1}{2}\sum_s \tilde{K}_{ii}(t,s)\chi_{ii}(t,s)\right] + \sum_i \log\,\tilde{Z}_i
\end{aligned}
\tag{11}
$$

Here $\langle \circ \rangle_D$ means taking an average with respect to a distribution $D$ (here $P_{\Gamma}$). Thus the minimum $Q_{\Gamma}$ satisfies:

$$\frac{\partial F}{\partial \tilde{h}_i(t)} = -M_i(t) + \frac{\partial\,\log\,\tilde{Z}_i}{\partial \tilde{h}_i(t)} = -M_i(t) + \tilde{M}_i(t) = 0 \tag{12}$$

$$\frac{\partial F}{\partial \tilde{K}_{ii}(t,s)} = -\chi_{ii}(t,s) + \frac{\partial\,\log\,\tilde{Z}_i}{\partial \tilde{K}_{ii}(t,s)} = -\chi_{ii}(t,s) + \tilde{\chi}_{ii}(t,s) = 0 \tag{13}$$

Here the right equality comes from the properties the partition function. Thus, the Uncoupled constraints (denoted with $\sim$) exactly match the true constraints, $M_i(t)$ and $\chi_{ii}(t,s)$.

 

**Reverse.**

$$
\begin{aligned}
R(\{\tilde{h}_i(t)\}, \{\tilde{K}_{ii}(t,s)\}) &= D_{\mathrm{KL}}(Q_\Gamma \,||\, P_\Gamma) \\
&= \langle \log\, Q_\Gamma \rangle_Q - \langle \log\, P_\Gamma \rangle_Q \\
&= \sum_{i,t}\left[\tilde{h}_i(t) - h_i(t)\right]\tilde{M}_i(t) + \frac{1}{2}\sum_{i,t,s}\left[\tilde{K}_{ii}(t,s) - K_{ii}(t,s)\right]\tilde{\chi}_{ii}(t,s) \quad (14) \\
&\quad - \frac{1}{2}\sum_{i,t,s}\sum_{j\neq i}K_{ij}(t,s)\tilde{M}_i(t)\tilde{M}_j(s) + \log\, Z - \sum_i \log\, \tilde{Z}_i
\end{aligned}
$$

Now because we have a unique mapping between our Lagrange multipliers and our constraints, $(\{\tilde{h}_i(t)\}, \{\tilde{K}_{ii}(t,s)\}) \leftrightarrow (\{\tilde{M}_i(t)\}, \{\tilde{\chi}_{ii}(t,s)\})$, we can find the minimum of the KL divergence in two different ways: we can either keep the Lagrange multipliers fixed and toggle the constraints or the other way around. Here we readily arrive at this minimum by choosing the former:

$$
\frac{\partial R}{\partial \tilde{M}_i(t)} = \tilde{h}_i(t) - h_i(t) - \sum_s \sum_{j\neq i} K_{ij}(t,s)\tilde{M}_j(s) = 0, \qquad \frac{\partial R}{\partial \tilde{\chi}_{ii}(t,s)} = \tilde{K}_{ii}(t,s) - K_{ii}(t,s) = 0 \quad (15)
$$

Collectively, these equations give the mean-field relations Eq 6.

## Linear Response Theory

Here we show how to estimate the pairwise interactions, $K_{ij}(t,s)$ using our Linear Coupling Approximation. Our approach naturally follows from Linear Response Theory [35]. We first recognize that, from the properties of Maximum Caliber distributions, $\frac{\partial M_i(t)}{\partial h_j(s)} = C_{ij}(t,s)$. Thus from Eq 6:

$$
\begin{aligned}
\frac{\partial h_i(t)}{\partial M_j(s)} = (C^{-1})_{ij}(t,s) &\approx \frac{\partial \tilde{h}_i(t)}{\partial M_j(s)} - K_{ij}(t,s) \\
&= -K_{ij}(t,s), \quad i \neq j
\end{aligned} \qquad (16)
$$

Since we already have estimates for our single trajectory Lagrange multipliers, we only use Eq 16 for the pairwise terms $i \neq j$; since our Uncoupled estimates depend on single-trajectory ($i = j$) terms only, their derivative is 0. Here the relationship is approximate because we neglect the derivatives of $K$ with respect to $M$; we assume that these terms are small, but their inclusion would lead to higher order corrections [61]. Due to our Linear Coupling Approximation, our couplings are only approximate, $K'$. These results directly imply Eq 7.

## Quantifying the accuracy of Linear Coupling

Here we compute the exact error of our Linear Coupling Approximation for an analytically solvable, but general model system. In particular, we follow the activities of two dynamically correlated agents, A and B. The activities of these agents, given by $v_1(t)$ and $v_2(t)$, are normally distributed and stationary (i.e. the Max Cal distribution given the vector of means $\vec{M}$ and the matrix of covariances **C** of our agent activities). Given the nature of normal distributions, it is possible to determine the *exact* Lagrange multipliers $\vec{h}$ and **K**:

$$
\mathbf{K} = -\mathbf{C}^{-1}, \qquad \vec{h} = -\mathbf{K}\vec{M} \qquad (17)
$$

 

Without any loss of generality, we set the means $\vec{M}$ (and hence the Lagrange multipliers $\vec{h}$) to 0 and focus our interest on inferring the couplings $\mathbf{K}$. To simplify our analysis, we first rewrite the covariance matrix $\mathbf{C}$ in terms of the auto-covariance matrix $\mathbf{C}_A$ of each agent and their cross-covariance $\mathbf{C}_C$:

$$\mathbf{C} = \begin{pmatrix} \mathbf{C}_A & \mathbf{C}_C \\ \mathbf{C}_C & \mathbf{C}_A \end{pmatrix} \tag{18}$$

To evaluate the accuracy of our Linear Coupling Approximation, we use Eqs 6 and 7 to compute the approximate couplings $\mathbf{K}'$. The first step in this process is solving the associated uncoupled problem. This is equivalent to solving for the couplings when cross-covariances between agents ($\mathbf{C}_C$) are ignored. I.e.:

$$\tilde{\mathbf{K}} = -\tilde{\mathbf{C}}, \qquad \tilde{\mathbf{C}} = \begin{pmatrix} \mathbf{C}_A & \mathbf{0} \\ \mathbf{0} & \mathbf{C}_A \end{pmatrix} \tag{19}$$

Where $\sim$ is used to represent our Uncoupling Approximation. Thus, from Eq 6, $\mathbf{K}'_A = \tilde{\mathbf{K}}_A = -\mathbf{C}_A^{-1}$. Finally, to find $\mathbf{K}'_C$, we need to compute the full inverse matrix $\mathbf{C}^{-1}$. Using standard results from linear algebra, we find that $\mathbf{K}'_C$ (the negative of the off-diagonal of this inverse matrix) is given by:

$$\mathbf{K}'_C = \mathbf{C}_A^{-1} \mathbf{C}_C \mathbf{C}_A^{-1} \tag{20}$$

Thus our final estimates for the couplings are given by:

$$\mathbf{K}' = \begin{pmatrix} \mathbf{K}'_A & \mathbf{K}'_C \\ \mathbf{K}'_C & \mathbf{K}'_A \end{pmatrix} = \begin{pmatrix} -\mathbf{C}_A^{-1} & \mathbf{C}_A^{-1} \mathbf{C}_C \mathbf{C}_A^{-1} \\ \mathbf{C}_A^{-1} \mathbf{C}_C \mathbf{C}_A^{-1} & -\mathbf{C}_A^{-1} \end{pmatrix} \tag{21}$$

To evaluate the accuracy of our Linear Coupling estimates, we invert Eq 21 and compute $\mathbf{C}'$:

$$\mathbf{C}' = -(\mathbf{K}')^{-1} = \begin{pmatrix} \mathbf{C}_A^{-1} & -\mathbf{C}_A^{-1} \mathbf{C}_C \mathbf{C}_A^{-1} \\ -\mathbf{C}_A^{-1} \mathbf{C}_C \mathbf{C}_A^{-1} & \mathbf{C}_A^{-1} \end{pmatrix}^{-1} = (\mathbf{I} - \mathbf{B}^2)^{-1} \mathbf{C} \approx (\mathbf{I} + \mathbf{B}^2) \mathbf{C}$$
$$\mathbf{B} = \begin{pmatrix} \mathbf{C}_A^{-1} \mathbf{C}_C & \mathbf{0} \\ \mathbf{0} & \mathbf{C}_A^{-1} \mathbf{C}_C \end{pmatrix} \tag{22}$$

Here the approximation in Eq 22 comes from the geometric relation $(1 - x)^{-1} \approx 1 + x$. To guarantee that the error in $\mathbf{C}'$ is small, we need the eigenvalues of $\mathbf{B}$ to all be less than unity in magnitude (i.e. auto-correlations are stronger than cross-correlations). Here we quantify this error using the matrix 2-norm ($\|\circ\|$, or the magnitude of a matrix's largest eigenvalue. In particular, using $\alpha$ to denote the upper bound on the relative error between our approximation and the ground truth, we have that:

$$\frac{\|\mathbf{C}' - \mathbf{C}\|}{\|\mathbf{C}\|} = \frac{\|\mathbf{B}^2 \mathbf{C}\|}{\|\mathbf{C}\|} \leq \|\mathbf{B}\|^2 = \alpha \tag{23}$$

As an example, if the largest eigenvalue of $\mathbf{B}$ is 0.5 (self-interactions are roughly twice as strong as opposite interactions), our error is already guaranteed to be less than 25%. In general, the error decreases quadratically with shrinking cross-correlation strength. And while

perturbation theories, by nature, are often very accurate away from their predicted regions of convergence, here we also have a guaranteed bound on the error of our approach when inferring the dynamics of weakly-correlated agents.

## Toggle switch

**Uncoupling Approximation.**   Here we derive analytical relations for the key features (criticality and bistability) of the genetic toggle switch using our Uncoupled (mean-field) approach. First, the full Max Cal distribution for this system [24] is given by:

$$P(l_A, l_B, l_\alpha, l_\beta) = Z^{-1}(N_A, N_B) \binom{N_A}{l_A} \binom{N_B}{l_B} e^{h_P[l_\alpha + l_\beta] + h_S[l_A + l_B] + K[l_A l_\beta + l_B l_\alpha]} \qquad (24)$$

Here the partition function $Z$ depends on the protein numbers $N_A$ and $N_B$ at the beginning of each $\delta t$ interval. We next use our Uncoupled approach to find an approximate analytical form for $Z$ (and thus our trajectory probabilities). Thus we want to find effective production and survival rates $\tilde{h}_{P,A}, \tilde{h}_{S,A}, \tilde{h}_{P,B}$, and $\tilde{h}_{S,B}$ such that $A$ and $B$ can be treated independently. From Eq 6, the effective fields are given by:

$$\begin{aligned} \tilde{h}_{S,A} = h_S + K\langle l_\beta \rangle, \quad \tilde{h}_{P,A} = h_P + K\langle l_B \rangle \\ \tilde{h}_{S,B} = h_S + K\langle l_\alpha \rangle, \quad \tilde{h}_{P,B} = h_P + K\langle l_A \rangle \end{aligned} \qquad (25)$$

By symmetry, we focus only on the equations for protein $A$. Here the Uncoupled distribution $Q_A$ is given by:

$$Q_A(l_A, l_\alpha) = Z_A^{-1}(N_A) \binom{N_A}{l_A} e^{\tilde{h}_{P,A} l_\alpha + \tilde{h}_{S,A} l_A} \qquad (26)$$

Conveniently, the Uncoupled partition function $Z_A$ has a closed form (SI in [24]:

$$\begin{aligned} Z_A(N_A) \quad &= (1 + e^{\tilde{h}_{P,A}})(1 + e^{\tilde{h}_{S,A}})^{N_A} \\ &\approx e^{\tilde{h}_{S,A} N_A} + e^{\tilde{h}_{P,A} + \tilde{h}_{S,A} N_A} + N_A e^{\tilde{h}_{S,A}(N_A - 1)} \end{aligned} \qquad (27)$$

with an analogous equation for $Z_B$. Here assumed (for simplicity) that since $\delta t$ is small, maximally one reaction will happen (either degradation or production) per time interval.

Additionally, the master equation, as well as the stationary distribution are well-known for the Uncoupled system (SI Eq 7 in [24]). For this process, the stationary distribution is a Poisson distribution with mean $\langle N_A \rangle$. Here $\langle N_A \rangle$ is always given by a stable point of the system (which one depends on which state $A$ is in).

**Finding the critical point.**   We next show how to use these equations to understand the critical transition of the genetic toggle switch. We can do this by examining the stationary points, or the $(N_A, N_B)$ pairs where the average production and degradation of both species are equal. A key property of partition functions, such as $Z_A$, is that we can compute averages over our model quantities ($l_\alpha$ and $l_A$) directly from the derivatives of these functions. In particular, we can directly find the points where production and degradation are equal:

$$\langle l_\alpha \rangle + \langle l_A \rangle = \frac{\partial \log Z_A}{\partial \tilde{h}_{P,A}} + \frac{\partial \log Z_A}{\partial \tilde{h}_{S,A}} = N_A + Z_A^{-1}(N_A) e^{\tilde{h}_{S,A} N_A} \left[ e^{\tilde{h}_{P,A}} - N_A e^{-\tilde{h}_{S,A}} \right] \qquad (28)$$

We can then think of the bracketed term (which we rewrite slightly for later convenience) as analogous to a force:

$$F_A = e^{\tilde{h}_{P,A} + \tilde{h}_{S,A}} - N_A \tag{29}$$

Here when this force is positive (production is greater than degradation), $N_A$ is likely to increase. Likewise, when the force is negative, the opposite is true. When they are equal, $N_A$ is a stationary point and the force is 0. These are the points where $N_A$ is equal to the average number of proteins $A$ produced minus the number degraded:

$$\langle l_\alpha \rangle + \langle l_A \rangle = N_A \tag{30}$$

Now from Eq 25, we have that the stationary points also satisfy

$$N_A = e^{\tilde{h}_{P,A} + \tilde{h}_{S,A}} = \Lambda e^{K(\langle l_\beta \rangle + \langle l_B \rangle)} \tag{31}$$

where $\Lambda = e^{h_P + h_S}$. Combining this with Eq 30 (and analogous equations for $B$), we find that the stationary points satisfy the coupled equations:

$$N_A = \Lambda e^{K N_B}, \quad N_B = \Lambda e^{K N_A} \tag{32}$$

Next we ask when these points are stable and when they are unstable. To do this, we evaluate how the force, $F_A$ changes as we toggle $N_A$ away from the fixed point. Using Eqs 29 and 32:

$$\frac{dF_A}{dN_A} = [\Lambda e^{K N_B} - N_A]' = K N_B \frac{dN_B}{dN_A} - 1 = K^2 N_B N_A - 1 \tag{33}$$

Thus a stationary point $(N_A, N_B)$ is stable when $K^2 N_A N_B > 1$ and unstable when $K^2 N_A N_B < 1$. As $K$ changes, so might the stability of a fixed point. In particular, as we vary $K$ the fixed point corresponding to coexistence of both proteins ($N_A = N_B = N_0$) changes from unstable to stable. This change occurs at the critical point: $K_c^2 N_0^2 = 1$. Since $N_0$ has to be positive and $K$ is negative or 0. Thus, from Eq 32,

$$K_c = -\frac{1}{N_0} \Rightarrow -\frac{1}{K_c} = \Lambda e^{-1} \Rightarrow K_c = -e^{1 - h_P - h_S} \tag{34}$$

## Neural network

**Selecting the Ground Truth model.**   Here we provide additional mathematical details of our method. In particular, we discuss how we chose our Ground Truth, brain model and how we, in practice, back-infer the dynamical couplings between our synthetic network of neurons. We chose our couplings to capture the key properties of the experimental observations described for static [47] and dynamic [27] clusters of real neurons. First, the heterogeneity of neural interactions ($i \neq j$) can be captured by choosing $K_{ij}(\tau)$ ($\tau = |t - s|$) from a normal distribution with mean ($K_0 a^{-\tau}$) and standard deviation ($K_\delta a^{-\tau}$) [47]. For simplicity, we choose $K_0 = K_\delta$. Here $a > 1$ describes the rate that correlations between neurons decay with time. Second, in weakly-interacting systems, such as networks of neurons, self-interactions ($i = j$) are much stronger than pair-interactions ($i \neq j$). Except at $\tau = 0$ (since $K_{ii}(0) = 0$ for the Ising model), we choose, without loss of generality, $K_{ii}(\tau) = 20 K_0 a^{-\tau}$. Third, since neurons have a strong tendency towards silence, we chose $h_i(t) = h_0$ ($h_0 < 0$) for all neurons. Fourth and most importantly, experimentally observed, neuronal avalanches can only occur when pairwise couplings are tuned near a critical point [27]; below this point, neural activity is uncorrelated and

random, while below this point it is strongly correlated and perpetually silent. Up to a change of scale, we can choose $K_0$ = .015 and $a$ = 4 for our convenience. To tune our network near criticality, we choose $h_0$ = −.1; here the very weak correlations between our synthetic neurons ($\approx$ .02 on average) can occasionally sum together to create a cascade of neural activation ("avalanche").

**Implementation details.**   Here we describe how we computed our Linear Coupling estimates of the Ground Truth Lagrange multipliers described for our toy brain example. First, we used a standard Metropolis-Hastings Markov Chain Monte Carlo (MCMC) algorithm ($5 \times 10^5$ iterations) to compute the means and correlations between our synthetic neurons. From these constraints, we computed our estimates for the pairwise couplings, $K_{ij}(\tau)$ using Eq 7. When $\tau \geq 4$, couplings are, on average, greater than $4^4 \approx 100$ fold weaker than at $\tau = 0$ and were safely neglected. Second, each of the Uncoupled problems is simply a 4-spin Ising model, constrained by the Ground Truth means and autocorrelations, and was solved exactly for each of our $N$ = 40 synthetic neurons. Finally, Eq 6 was used to reconstruct the remaining Ground Truth Lagrange multipliers from our Uncoupled estimates.

## Author Contributions

**Conceptualization:** Corey Weistuch.

**Formal analysis:** Corey Weistuch.

**Funding acquisition:** Lilianne R. Mujica-Parodi, Ken A. Dill.

**Methodology:** Corey Weistuch, Luca Agozzino, Ken A. Dill.

**Validation:** Corey Weistuch.

**Visualization:** Corey Weistuch, Luca Agozzino.

**Writing – original draft:** Corey Weistuch, Luca Agozzino, Lilianne R. Mujica-Parodi, Ken A. Dill.

**Writing – review & editing:** Corey Weistuch, Luca Agozzino, Lilianne R. Mujica-Parodi, Ken A. Dill.

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
