## [Decision Letter · Decision Letter 0]

9 Jul 2020

Dear Dr. Dill,

Thank you very much for submitting your manuscript "Inferring a network from dynamical signals at its nodes" for consideration at PLOS Computational Biology.

As with all papers reviewed by the journal, your manuscript was reviewed by members of the editorial board and by several independent reviewers. In light of the reviews (below this email), we would like to invite the resubmission of a significantly-revised version that takes into account the reviewers' comments.

As you shall see in the reviewers' comments, all reviewers find the work interesting and results overall convincing. The key issues to address are on the rationale for choosing the method (Max Cal), the intuition on why the method works well for specific examples, and the broad applicability of the method.

We cannot make any decision about publication until we have seen the revised manuscript and your response to the reviewers' comments. Your revised manuscript is also likely to be sent to reviewers for further evaluation.

Sincerely,

Lingchong You

Associate Editor

PLOS Computational Biology

Jason Haugh

Deputy Editor

PLOS Computational Biology

Reviewer's Responses to Questions

**Comments to the Authors:**

Reviewer #1: Weistuch et al provide an approximate solution to inferring network structure given temporal signal strength using Maximum Caliber. The authors show that this approach can infer the appropriate network constraints of the classic Collins Toggle switch, and a more complex brain neural network. Overall, the idea is interesting and the results are compelling. However, the use of network inference, such as Bayesian-type and ML inference models for inferring optimal network topologies is not a new notion on its own, and the authors did not sufficiently cast their results in the context of the (very widely established) field. Therefore, it was unclear how, and under what conditions, this approach performs comparably/superior etc. The authors should both elaborate with text, and use simulations, to contrast the advantages and/or utility of their approach to other established ones; this will better elucidate the overall constraints on performance that were not sufficiently described or explored. Moreover, given the abundance of available temporal scientific data and known network topologies, the authors should demonstrate the accuracy of this method using actual experimental data, rather than entirely simulated experiments. Without that, it’s difficult to assess the capabilities and limitations of the author’s approach compared to other’s that have been tested and previously implemented. Moreover, this will elucidate the range of experimental variability that it can sustain, which currently is unclear in the manuscript.

Reviewer #2: Please see attached for my review.

Reviewer #3: The authors present approximate solutions to a challenging inverse problem -- inferring interactions between agents from time-dependent signals at individual agents. Using the Principle of Maximum Caliber (Max Cal) - the dynamic analogue of Maximum Entropy inference - the authors make two leading-order approximations – a mean-field solution (decoupled nodes with effective parameters) and a linear-coupling solution (with linear response to perturbations around the mean-field solution). The authors then compared the approximate solutions to simulated data in two specific systems of broad interests: a genetic toggle switch and a network of neurons. They demonstrated that the mean-field solution (uncoupled Max Cal) captures the bifurcation diagram underlying the transition to a bistable toggle switch, and that linear-coupling is necessary for capturing neural avalanches and the variance of activity. The presented approach is intuitive and allows order-by-order inference of the coupling effects among nodes in a network. The resulting reduction in dimensionality represents a promising first step toward analysis, at enhanced computational efficiency, of large networks with complex feedback and nonlinearities.

Specific points:

1. Since the audience might not be familiar with the insight and strength of Max Cal, they would benefit from a concise review of the status of the field, past applications of Max Cal, why it is the method of choice for inferring dynamic processes, and what are the potential limitations, in the Introduction. This would help motivate the development of approximate approaches and establish the novelty of this work.

2. To demonstrate the value of this new development, it would be very helpful to show how the performance of approximate Max Cal compares to alternative approaches for inferring interactions in a network. Such comparison would help the reader to appreciate why and when this approximate approach is a good choice.

3. Can the authors quantify in some way the regime of applicability of the approximate solutions? In other words, under what conditions (e.g. general features of the network) shall we expect the approximations to be quantitatively useful?

4. Since this work sets out to address the challenge of high dimensionality of the inference problem, it would be more convincing if the result can be demonstrated for a larger system, e.g. a genetic regulatory network with coupled modules. How would the approximate solutions perform in cases of more complex nonlinearities and feedback than those in a single toggle switch?

5. Can the authors comment on the broader insights gained from these two particular examples? E.g. why can the uncoupled solution capture the critical point and full distributions away from it?

6. To obtain the dynamic mean field equations in the linear coupling approximation, reverse KL divergence between the true Max Cal distribution and the approximate one is minimized. Can the authors clarify why “the greatest insight comes from choosing the former” (i.e. minimizing with respect to the constraints while keeping the Lagrangian multipliers fixed)? Also, is there an intuitive way to see that, to the leading order, solutions from minimizing the forward and reverse KL divergence would match?

7. A minor point: in SI equations B2 and B3, should the first term be entropy (S instead of C)?

**Have all data underlying the figures and results presented in the manuscript been provided?**

Reviewer #1: Yes

Reviewer #2: Yes

Reviewer #3: Yes

PLOS authors have the option to publish the peer review history of their article (what does this mean?). If published, this will include your full peer review and any attached files.

Reviewer #1: No

Reviewer #2: **Yes: **Hyun Youk

Reviewer #3: No
---

## [Decision Letter · Decision Letter 1]

12 Oct 2020

Dear Dr. Dill,

We are pleased to inform you that your manuscript 'Inferring a network from dynamical signals at its nodes' has been provisionally accepted for publication in PLOS Computational Biology.

Best regards,

Lingchong You

Associate Editor

PLOS Computational Biology

Jason Haugh

Deputy Editor

PLOS Computational Biology

Reviewer's Responses to Questions

**Comments to the Authors:**

Reviewer #1: The authors have clarified my main questions adequately

Reviewer #2: The authors have addressed all my comments.

Reviewer #3: The authors have addressed my comments and suggestions.

**Have all data underlying the figures and results presented in the manuscript been provided?**

Reviewer #1: Yes

Reviewer #2: Yes

Reviewer #3: Yes

PLOS authors have the option to publish the peer review history of their article (what does this mean?). If published, this will include your full peer review and any attached files.

Reviewer #1: No

Reviewer #2: No

Reviewer #3: No

---

## [Editor Report · Acceptance letter]

16 Nov 2020

PCOMPBIOL-D-20-00777R1 

Inferring a network from dynamical signals at its nodes

Dear Dr Dill,

I am pleased to inform you that your manuscript has been formally accepted for publication in PLOS Computational Biology. Your manuscript is now with our production department and you will be notified of the publication date in due course.

With kind regards,

Nicola Davies
